# Nutritional Targeting of the Microbiome as Potential Therapy for Malnutrition and Chronic Inflammation

**DOI:** 10.3390/nu12103032

**Published:** 2020-10-03

**Authors:** Lena Schröder, Sina Kaiser, Burkhardt Flemer, Jacob Hamm, Finn Hinrichsen, Dora Bordoni, Philip Rosenstiel, Felix Sommer

**Affiliations:** Institute of Clinical Molecular Biology, University of Kiel, Rosalind-Franklin-Straße 12, 24105 Kiel, Germany; l.schroeder@ikmb.uni-kiel.de (L.S.); s.kaiser@ikmb.uni-kiel.de (S.K.); b.flemer@ikmb.uni-kiel.de (B.F.); j.hamm@ikmb.uni-kiel.de (J.H.); f.hinrichsen@ikmb.uni-kiel.de (F.H.); d.bordoni@ikmb.uni-kiel.de (D.B.); p.rosenstiel@mucosa.de (P.R.)

**Keywords:** nutritional intervention, microbiome, immunometabolism, SCFA, malnutrition

## Abstract

Homeostatic interactions with the microbiome are central for a healthy human physiology and nutrition is the main driving force shaping the microbiome. In the past decade, a wealth of preclinical studies mainly using gnotobiotic animal models demonstrated that malnutrition and chronic inflammation stress these homeostatic interactions and various microbial species and their metabolites or metabolic activities have been associated with disease. For example, the dysregulation of the bacterial metabolism of dietary tryptophan promotes an inflammatory environment and susceptibility to pathogenic infection. Clinical studies have now begun to evaluate the therapeutic potential of nutritional and probiotic interventions in malnutrition and chronic inflammation to ameliorate disease symptoms or even prevent pathogenesis. Here, we therefore summarize the recent progress in this field and propose to move further towards the nutritional targeting of the microbiome for malnutrition and chronic inflammation.

## 1. Introduction

The diverse assembly of microorganisms in the gastrointestinal tract referred to as gut microbiome largely contributes to human physiology, for example by producing essential vitamins, educating the host’s immune system, creating an ecological barrier against pathogenic infection or the generation of dietary metabolites that serve as a main energy source for colonic epithelial cells [1,2]. Disturbances in the homeostatic interactions with the microbiome lead to dysbiosis, which has been associated with several diseases ranging from obesity, diabetes, chronic inflammation and cancer to neurological disorders [3]. Host genetics but also and mainly environmental factors shape the composition and function of the microbiome [4]. These environmental factors include delivery mode, housing conditions, hygiene, medication, antibiotics and nutrition [5]. Of all these factors, nutrition has the most dominant effect on the microbiome. Malnutrition refers to all forms of imbalanced energy and/or nutrient intake [6] and therefore manifests as undernutrition, micro-nutrient-related malnutrition or overnutrition leading to overweight and obesity. Both under- and overnutrition represent major global health and socioeconomic challenges. Undernutrition is mainly prevalent in developing countries while overnutrition is mainly associated with Western societies [7]. Nutritional interventions could be a sustainable and cost-effective way to save lives and reduce healthcare costs. However, it has become evident that the long-term success of these nutritional interventions greatly varies between individuals. Success depends on whether the dietary recommendations are practicable for each individual and whether these measures indeed yield the intended physiological changes. Given the central role of the microbiome for the host’s physiology and its plasticity towards dietary input, these individualized responses to nutritional interventions seem to, at least in part, depend on the subject’s microbiome. Recent studies indeed substantiated this link [8,9,10] and therefore, we propose to analyze and to incorporate each individual’s microbiome when designing future nutritional interventions.

### 1.1. Malnutrition and the Microbiome

Severe undernutrition leads to impaired growth (stunting), neurodevelopmental abnormalities and immune dysfunction [9,11]. Commonly used anthropometric measures to quantify malnutrition severity include, for example, the ratios of weight-for-height (to indicate wasting), height-for-age (stunting) and weight-for-age (underweight) [12]. Additionally, malnutrition is associated with diarrhea and intestinal inflammation and mortality is greatly increased among undernourished children [7]. Traditionally, the pathogenesis of undernutrition has solely been attributed to a lack of dietary energy intake [13]. More recently, the gut microbiome has been identified as a central factor in undernutrition because of its ability to harvest nutrients from the diet [14]. Dysbiotic changes were identified in the fecal microbiomes of undernourished children in Bangladesh [15] and Malawi [16], which led to immature configurations that were similar to that of healthy but chronologically younger children, thereby linking the immaturity of the microbiome with undernutrition. Microbiome immaturity and reduced bacterial diversity were associated with reduced measures of anthropometry whereas increased fecal *Acidaminococcus* abundance was associated with future growth deficits [7]. Experiments employing gnotobiotic mice indeed demonstrated a causal role for the microbiome in disease pathogenesis. The transfer of the microbiome of undernourished children to germ-free mice via fecal microbiome transfer (FMT) also transferred the undernourished phenotype [16,17]. The international standard therapeutic approach for acute malnutrition is to supply access to an improved nutrition via ready-to-use therapeutic food (RUTF), which increases body weight and reduces the mortality in affected children [9]. However, the malnutrition-induced phenotypic defects are only partially normalized by RUTF in a fraction of recipients [18,19], which indicates highly personalized responses that potentially involve the gut microbiome. The lack of long-term success of RUTF therapy could be associated with an only transient maturation of the gut microbiome [16,17]. Clinical trials using a combination of antibiotics together with RUTF revealed lower mortality rates and improved weight gain compared to RUTF treatment alone [20]. Recently, candidate food supplements, termed microbiome-directed complementary food (MDCF), have been developed in pre-clinical models and indeed restored microbiome composition and function but also increased biomarkers of normal growth [9]. Furthermore, sialylated milk oligosaccharides were identified as depleted in undernourished children from Malawi and providing these compounds to gnotobiotic mice and piglets augmented growth in a microbiome-dependent manner in both experimental models [21]. Another way for intestinal bacteria to foster growth may be via fueling into the host’s lipid metabolism [22,23]. Recently, a healthy gut microbiome composition has even been linked to an adequate response to oral vaccination with cholera toxin in undernutrition [24].

Overnutrition, the other form of malnutrition, results from a higher energy intake than caloric expenditure thereby leading to the build-up of adipose tissue [25]. Despite the energetic surplus, overnutrition is also often accompanied by micronutrient malnutrition due to an unbalanced diet rich in saturated fat, simple sugars, salt and processed foods but low in protein, fruits, fiber and complex carbohydrates [26]. The microbiome contributes to the pathophysiology and development of obesity [27]. Germ-free mice were resistant to diet-induced obesity when fed a high-fat, high-sugar diet [28]. Genetically obese mice and mice fed a Western diet but also obese humans harbor an altered microbiome [29,30] compared to lean healthy controls with an expansion of Firmicutes and simultaneous reduction in Bacteroidetes. Long-term exposure to a Western diet even induced severe reductions in the species complexity of the microbiome with extinctions of entire phylogenetic groups leading to compromised microbiome function [31], which argues for probiotic approaches to restore normal microbiome complexity. The transfer of the microbiome from a lean donor to an obese recipient via FMT led to improved insulin sensitivity in obese humans [32,33] but does not seem to trigger persistent weight loss [34]. Moreover, the microbiome also plays a role in energy extraction from the diet. Feces from mice harboring a microbiome contained less energy than those of germ-free mice [28]. The transfer of the fecal microbiome from obese to germ-free mice led to a greater increase in fat mass compared to germ-free mice that were colonized with the microbiome of lean mice [35]. The microbiome contributes to energy extraction from the diet, for example, through the fermentation of otherwise non-digestible fibers and resistant starches. Central products of these microbial metabolic activities are short-chain fatty acids (SCFAs) such as acetate, propionate and butyrate [36]. SCFAs are the major energy source for colonic epithelial cells and could potentially promote adiposity by contributing additional calories and by fueling into gluconeogenesis and lipogenesis in adipocytes or hepatocytes [37,38]. However, SCFAs also have many beneficial effects on host physiology and therefore targeting SCFAs has also been suggested as a potential therapeutic approach for obesity [39]. SCFAs play an important role in host immunity [40]. A reduction in fecal SCFAs has been linked to inflammatory and metabolic diseases such as obesity, diabetes and inflammatory bowel disease (IBD) [41]. Butyrate in particular possesses anti-cancer and anti-inflammatory properties [42].

### 1.2. Inflammation Dysregulates the Metabolic Interplay with the Microbiome

Chronic inflammation poses a serious stress to the affected organism, not only due to the inflammatory response itself but also through the altered nutritional status both from dysregulated metabolic activities of the host and interactions with the microbiome. It is well established that inflammation leads to dysbiosis and that a dysbiotic microbiome can in turn trigger inflammatory responses. For example, undernutrition often associates with environmental enteropathy, a chronic inflammation of the small intestine, which leads to profound changes in the microbiome and intraepithelial lymphocyte composition [43]. The transfer of IgA-targeted bacterial taxa from undernourished Malawian children induced environmental enteropathy in recipient gnotobiotic mice, which could be rescued by the administration of IgA-targeted bacteria from a healthy donor [44]. In obesity, the concept of endotoxin-mediated low-grade inflammation bridges nutrition, microbiome and inflammation. According to this theory, Western diet-induced dysbiosis increases the level of endotoxin, a component of the outer membrane of Gram-negative bacteria, which then reaches other organs such as the liver and adipose tissues via the blood circulation in higher concentrations that suffices to stimulate tissue-resident immune cells and thereby lead to immune responses and impaired organ functions [45]. In IBD, bacterial dysbiosis comprises a reduction in alpha diversity (the number of taxa) and in the abundance of Bacteroidetes, whereas the abundance of Actinobacteria and Proteobacteria are increased [3,46]. On a species level, these changes include an expansion of potential pathogens such as *Escherichia coli* whereas beneficial bacteria such as the butyrate-producing *Faecalibacterium prausnitzii* are reduced in IBD [47]. Metabolic analyses of inflamed mucosae revealed that central metabolites and metabolic pathways of both the host and the microbiome are affected by the inflammatory environment. For example, the decreased abundance of beneficial bacteria translated into reduced levels of SCFAs, especially butyrate, in the inflamed mucosa of IBD patients [48]. Furthermore, recent studies revealed that the production of butyrate by the microbiome also determines the responsiveness to biological therapy in IBD patients [49,50]. Supplementing diet with fermentable fiber (prebiotics) represents an attractive approach to foster the growth of beneficial bacteria and to restore missing metabolic functions, i.e., SCFA production. This can resolve inflammation or at least ameliorate metabolic stress caused by an inflammatory environment. Initial human studies indeed support the feasibility of this concept [10].

Another class of metabolites central to the host’s immune response are tryptophan metabolites. As for SCFAs, the levels of tryptophan metabolites are reduced in the serum of IBD patients [51]. Nutritional studies using gnotobiotic mice demonstrated that the lack of dietary tryptophan leads to colitis and the supplementation of tryptophan protected from inflammation [52]. Transfer of a dysbiotic microbiome from mice fed a tryptophan-deficient diet to germ-free recipients fed a normal diet was sufficient to cause colitis. In mice, the tryptophan-metabolizing strain *Peptostreptococcus russellii* protects from colitis [53] as it metabolizes tryptophan into indoleacrylic acid, the ligand of the aryl hydrocarbon receptor that functions by enhancing epithelial barrier function and reducing inflammatory responses. Dietary tryptophan also impacts epithelial immunity by boosting the production of antimicrobial peptides that regulate microbiome composition and protect from infection with opportunistic pathogens [52]. In contrast to IBD, the lack of dietary tryptophan protects from autoimmunity of the central nervous system in a murine model of multiple sclerosis due to impaired encephalitogenic T cell responses and profound alterations in the microbiome [54]. Notably, the protective effects of dietary tryptophan restriction were abrogated in germ-free mice and independent of prototypical sensors of tryptophan metabolites, indicating that these protective effects could be mediated via the metabolic functions of the microbiome. Other conditions such as obesity or diabetes also present a subclinical but chronic intestinal inflammation and therefore central concepts laid out here for nutritional manipulations of host–microbiome interactions in IBD are likely to apply across inflammatory diseases [55,56].

### 1.3. Nutritional Interventions—Where Do We Go from Here?

Findings from exploratory analyses and initial proof-of-concept studies using preclinical models of dietary interventions have begun to be transferred to the clinic. For example, a wealth of epidemiological studies have shown that the consumption of a Mediterranean diet rich in unsaturated fats is associated with an extended lifespan and a lower prevalence of diseases with chronic subclinical inflammation such as coronary heart disease or diabetes [57]. In a randomized and controlled trial with obese individuals, a Mediterranean diet increased the abundance of the SCFA-producing bacteria *Roseburia* and *Oscillospira*, whereas a diet rich in complex carbohydrates increased the abundance of *Prevotella* and *F. prausnitzii*. Both diets increased insulin sensitivity [58]. Similarly, deficiency in SCFA production has been associated with diabetes [59,60,61]. In a recent clinical trial with diabetic patients, the supplementation of an isoenergetic high-fiber diet modulated the abundance of several SCFA-producing bacteria to various degrees between individuals. In subjects with fiber-induced high levels of *Bifidobacterium pseudocatenulatum* C95 and *F. prausnitzii* CAG0106, the levels of hemoglobin A1c were reduced and the overall clinical outcome was improved [62]. Furthermore, a negative correlation has been reported between the fecal abundance of *Akkermansia muciniphila* and obesity, diabetes and hypertension [63,64,65,66,67]. A gnotobiotic mouse model demonstrated that unsaturated dietary lipids such as fish oil increased the fecal abundance of *A. muciniphila* [68]. A recent pilot study therefore tested the administration of *A. muciniphila* to obese, insulin-resistant patients. Indeed, dietary supplementation with *A. muciniphila* for 3 months reduced body weight, blood markers for liver dysfunction or inflammation and improved several metabolic parameters such as insulin sensitivity [69]. Reductions in the abundance of *A. muciniphila* have also been reported for IBD [70], making *Akkermansia* an attractive target for microbiome-directed therapies for these inflammatory diseases. To date, only very few probiotic products have demonstrated beneficial effects in randomized clinical studies. One of these is the “De Simone Formulation”, a mixture of eight bacteria (four strains of *Lactobacillus*, three strains of *Bifidobacterium* and one strain of *Streptococcus*). This formulation improves intestinal barrier function and ameliorates disease symptoms in atherosclerosis, irritable bowel syndrome, ulcerative colitis, antibiotic-associated diarrhea and radiation-induced enteritis [71]. In IBD, exclusive enteral nutrition (EEN), which involves an entirely liquid formula diet, has proven an effective therapeutic option to induce remission in active Crohn’s disease with an even higher efficacy in the induction of mucosal healing compared to steroids [72]. Notably, EEN altered the microbiome by increasing overall bacterial diversity and the abundances of *Clostridium symbiosum*, *C. ruminantium*, *C. hathewayi*, *Ruminococcus torques* and *R. gnavus*, all of which are known SCFA-producers [73].

The role of the microbiome has mainly been studied in Western diseases of civilization and recent reviews called attention to the potential of personalized nutrition and microbiome-directed therapies, for example, the use of probiotics or FMT in obesity to replenish species that were lost due to prolonged unbalanced diets and thereby to restore microbiome diversity and proper ecosystem functionality [74,75,76,77]. In recent years, the impact of the microbiome has also been highlighted regarding malnutrition in developing countries. The standard therapy for severe undernutrition, supplementation with RUTF, could not fully restore microbiome dysbiosis and disease symptoms in malnourished children. Therefore, Gehrig and colleagues investigated in a randomized and double blinded human trial whether targeting the immature microbiome of malnourished children by microbiome-directed complementary food could potentially reduce the long-lasting side effects of malnutrition [9]. Indeed, supplementation with microbiome-directed complementary food led to the normalization of anthropometric measures, a higher production of butyrate and tryptophan metabolites and the restoration of a mature microbiome. Thus, precision nutritional interventions targeting the microbiome could potentially reduce the negative long-term effects of malnutrition [9]. Taken together, recent translational efforts began to transfer findings from exploratory and preclinical studies into functional human trials targeting specific functional host–microbiome interactions. This translation to humans is pivotal, as preclinical research models such as gnotobiotic animals on the one hand are essential tools to uncover functional principles in nutrition–host–microbiome interactions. On the other hand, germ-free animals also have an immature immune system and display differences in important immune components and the composition of the microbiome, which limits their use as human surrogates. Nonetheless, an improved understanding of host–microbiome interactions from a variety of diseases led to the development of novel interventions that could improve traditional therapeutic concepts.

## 2. Conclusions

Interactions between nutrition, the microbiome and host determine crucial physiological processes of the human metaorganism including inflammatory responses, metabolic functions as well as disease susceptibility and pathogenesis (Figure 1). We have begun to identify nutritional components (e.g., fiber, complex carbohydrates, unsaturated fat), microbial taxa (e.g., *A. muciniphila*, *F. prausnitzii*, *Roseburia* spp.) and metabolites (e.g., SCFAs or tryptophan metabolites) that contribute to the pathogenesis of malnutrition and chronic inflammation. In contrast to the human metabolic potential, the microbiome shows great plasticity towards environmental factors, i.e., diet and nutrition. Thus, the composition and function of the microbiome can be targeted with highly specific and personalized nutritional supplementations. We therefore propose to further develop nutritional interventions as an additional or alternative therapy to conventional treatment. This, of course, requires the thorough and critical benchmarking of the efficacy of these nutritional and microbiome-directed interventions compared to conventional or combination therapies in controlled clinical human studies. Developing a general broad-spectrum diet high in fermentable fiber and unsaturated fat that improves physiological parameters and metabolic health would be an ideal goal, as it represents a cost-effective way to improve public and individual health. However, this might not be feasible given the high interindividual dietary preferences and variation in the gut microbiome but also host genetics and immunity. The solution may therefore be personalized nutrition. This novel avenue inherently depends on characterizing both the patient’s genome and microbiome, making it expensive and time consuming but also very promising. However, economical restraints in developing countries might impair the widespread usage of personalized malnutrition therapy. Global initiatives that financially support these efforts and developing cost-effective precision nutrition approaches will be a key requirement for the upcoming years. The mechanisms by which personalized nutrition operate are not fully understood and the close monitoring of the metabolic and microbial adaptations will be necessary. Furthermore, human nutritional trials are not easy to conduct as one either has to rely on the strict compliance of the participants or alternatively controlled and expensive studies where similarly prepared meals for every participant under a supervised environment are needed. Despite these potential pitfalls, we considered combinatorial therapeutic approaches targeting both the microbiome and nutritional deficits as a promising non-invasive therapeutic option for various inflammatory and metabolic diseases (Figure 2). The feasibility of this approach has already been demonstrated, for example as therapy for malnutrition as microbiome-directed complementary food supplements outperformed RUTF, which solely treats the calorie and macronutrient deficits but does not normalize dysbiosis [9]. Future controlled clinical studies will shed light on whether the initial promising findings will hold up in larger replication studies and other diseases.

## Figures and Tables

**Figure 1 nutrients-12-03032-f001:**
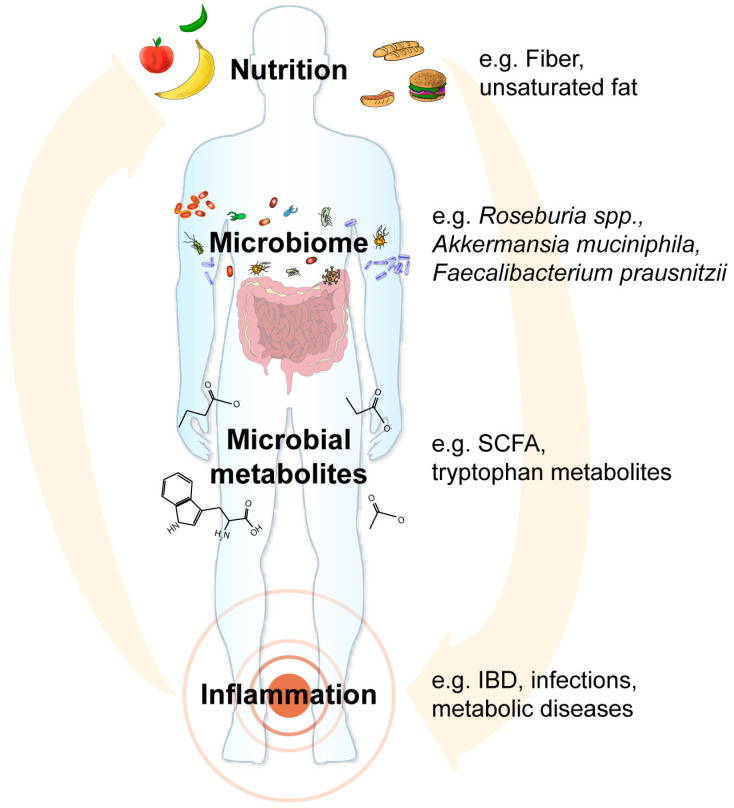
Concept of microbiome-directed nutritional interventions to treat malnutrition and chronic inflammation. Nutrition directly impacts the pathogenesis of malnutrition and inflammation, for example by fueling host metabolism. However, nutrition is also a main environmental factor shaping the composition and function of the microbiome. Detrimental microbiome alterations such as the loss of beneficial bacteria (for example *Akkermansia muciniphila*, *Faecalibacterium prausnitzii*, *Roseburia* spp.), which produce and supply short-chain fatty acids (SCFAs) and vitamins, or the expansion of pathobionts (for example Proteobacteria), which may cause infections in susceptible hosts, could also boost disease pathogenesis or might cause resistance to therapy approaches. In addition, microbial metabolites such as the SCFA butyrate or tryptophan metabolites control various physiological functions in the host ranging from inflammatory responses to the energy metabolism of epithelial cells. Therefore, microbiome-directed and personalized therapy approaches promise to improve treatment efficacy for malnutrition and chronic inflammation, for example inflammatory bowel disease (IBD).

**Figure 2 nutrients-12-03032-f002:**
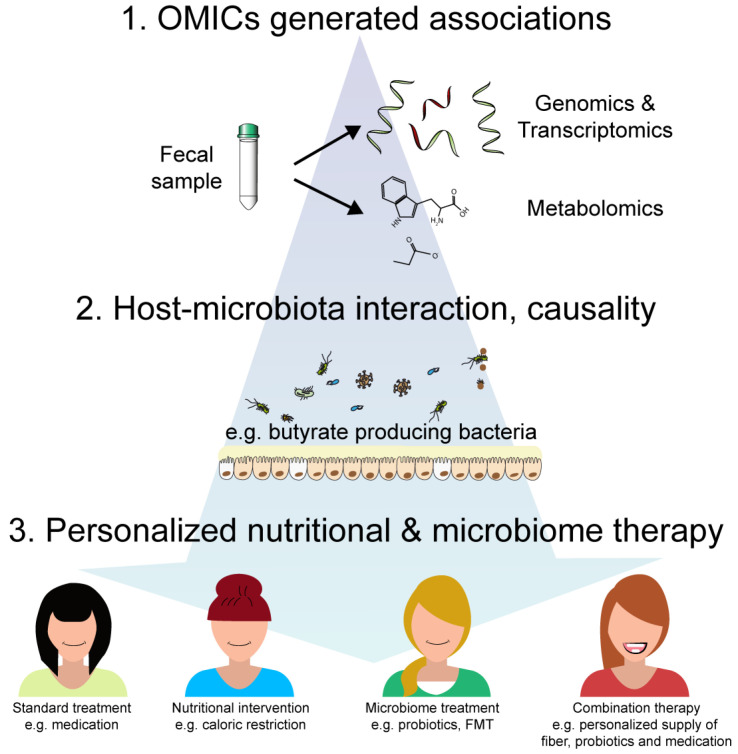
Roadmap for personalized therapy combining nutrition, microbiome and classical medication. In the past decade, OMICS studies based on next-generation sequencing or metabolomics revealed the associations of alterations in the genome, transcriptome or metabolome with disease states. Longitudinal studies allowed retrospective correlations of dysbiotic signatures with early, late or acute disease. Functional studies of host–microbiome interactions mainly using gnotobiotic animal model-established causality and identified, for example, the central role of the microbiome-related SCFAs and tryptophan metabolites for the regulation of inflammatory responses. Standard patient care (e.g., medication), nutritional interventions (e.g., caloric restriction) or microbiome-directed treatments (e.g., probiotics or fecal microbiome transfer (FMT)) all have beneficial physiological effects. However, combining these therapies in a personalized approach promises to yield a more effective and long-lasting treatment as it was already shown for microbiome-directed complementary food supplements, which outperformed ready-to-use therapeutic food (RUTF) in resolving malnutrition.

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
