# Peer review of "Nutritional Targeting of the Microbiome as Potential Therapy for Malnutrition and Chronic Inflammation"

_nutrients, 2020, doi:10.3390/nu12103032_

Round 1
Reviewer 1 Report
Overall Impression:This review manuscript covers a topic of important consideration. Readers of Nutrients journal are a good fit for individuals that will benefit from this manuscript. Topic is relevant to current interests of many biomedical researchers.
Comments-Good flow of concepts, well connected thoughts-The inclusion of both under and over nutrition in this discussion is important and the authors spent equal time discussing both and the implications of the microbiome in each case-Conclusion adequately discusses the challenges of creating a personalized nutritional intervention, though readers may appreciate suggested solutions to these obstacles (either already in use in other fields or new concepts). This is especially important in regards to under nutrition which occurs widely in areas with few resources. How canareas struggling with the most basic of healthcare needs seriously consider treating under nutrition with anything other than what is inexpensive and easily accessible?-A bit more detail about anthropometry measures and phenotypic defects described under “malnutrition and microbiome” would help.
Author Response
Reviewer 1
Overall Impression:
This review manuscript covers a topic of important consideration. Readers of Nutrients journal
are a good fit for individuals that will benefit from this manuscript. Topic is relevant to current
interests of many biomedical researchers.
We thank the reviewer for his/her kind words.
Comments:
-Good flow of concepts, well connected thoughts
We thank the reviewer for his/her kind words.
-The inclusion of both under and over nutrition in this discussion is important and the authors
spent equal time discussing both and the implications of the microbiome in each case
We thank the reviewer for his/her kind words.
-Conclusion adequately discusses the challenges of creating a personalized nutritional
intervention, though readers may appreciate suggested solutions to these obstacles (either
already in use in other fields or new concepts). This is especially important in regards to under
nutrition which occurs widely in areas with few resources. How can areas struggling with the
most basic of healthcare needs seriously consider treating under nutrition with anything other
than what is inexpensive and easily accessible?
We thank the reviewer for mentioning this important issue. We added some examples and a
brief discussion of possible limitations caused by economic restraints especially regarding
undernutrition.
The updated passage (lines 227-232) reads: “The solution may therefore be personalized
nutrition. This novel avenue inherently depends on characterizing both the patient's genome
and microbiome, making it expensive and time consuming but also very promising. Yet,
economical restraints in developing countries might impair widespread usage of personalized
malnutrition therapy. Global initiatives that financially support these efforts and to develop
cost-effective precision nutrition approaches will be a key requirement for the upcoming
years.”
-A bit more detail about anthropometry measures and phenotypic defects described under
“malnutrition and microbiome” would help.
We added a short summary of anthropometry measures and phenotypic defects for
malnutrition.
The updated passage (lines 48-51) reads: “Severe undernutrition leads to impaired growth
(stunting), neurodevelopmental abnormalities and immune dysfunction [9,11]. Commonly
used anthropometric measures to quantify malnutrition severity include, for example, the
ratios of weight-for-height (to indicate wasting), height -for-age (stunting) and weight-for-age
(underweight) [12].”

Reviewer 2 Report
The authors have highlighted several promising studies across a wide variety of diseases, and this is a nice overview of some newer results. However the concepts for obesity are underdeveloped in this report.
Microbiota supplementation in under-nutrition is well described, but the overnutiriton is not so simple. Beginning on line 78, The concept of energy extraction for a model of obesity is still controversial. Fermentation of SCFA is not accepted as the reasons for obesity; indeed replacement of SCFA is suggested as a therapy for obesity. This needs to be explained better in the manuscript.
Likewise the heterogeneity is an important concept in personal nutrition, as the authors point out, there will be an effect of genetics.
Host immunity is very important and this is not clear from lines 142 to 144. The endotoxin theory of inflammation and gut origin of obesity needs clarification, rather than linking this to IBD. Obesity and metainflammation is clearly an important clinical endpoint to treatment of diabetes and obesity and needs to be explained better.
The use of germ-free mice with an immature immune system and the limits of using this as a model for humans needs to be emphasized. The receptors for SCFA, TLR4, and immunopathology of these animals is unlike that of humans.
Author Response
Reviewer 2The authors have highlighted several promising studies across a wide variety of diseases, and this is a nice overview of some newer results. However the concepts for obesity are underdeveloped in this report.We thank the reviewer for his/her kind words.Microbiome dysbiosis and nutritional interventions in obesity gained significant attention in the past ten years leading to high impact original publications and also several recent reviews on this topic. We therefore focused more on the underrepresented topic of undernutrition,but we nonetheless mentioned overnutrition as complementary concept as well. We added a passagementioning these recent reviews.The updated passage(lines 187-191) reads:“The role of the microbiome has mainly been studied in western diseases of civilization and recent reviews called attention to the potential of personalized nutrition and microbiome-directed therapies, for example, the use of probiotics or FMT in obesity to replenish species that were lost due to prolonged unbalanced diets and thereby to restore microbiome diversity and proper ecosystem functionality [74-77]. In the past years, the impact of the microbiome has also been highlighted regarding malnutrition in developing countries.”Microbiota supplementation in under-nutrition is well described, but the overnutiriton is not so simple. Beginning on line 78, The concept of energy extraction for a model of obesity is still controversial. Fermentation of SCFA is not accepted as the reasons for obesity; indeed replacement of SCFA is suggested as a therapy for obesity. This needs to be explained better in the manuscript.The mechanisms involved inthe interactions between the microbiome andmalnutrition are mostly not understood up until today.We fully agree with the reviewer that SCFA per se do not function detrimental,but instead have many beneficial functions, as also laid out in our manuscript here. However, for obesity the energy surplus caused by improved energy extraction is a plausible hypothesisto develop adiposity.We clarified this passage.The updated passage(lines 96-106) reads:“The microbiome contributes to energy extraction from the diet, for example, through the fermentation of otherwise non-digestible fibers and resistant starches. Central products of these microbial metabolic activities are short-chain fatty acids (SCFA) such as acetate, propionate and butyrate [36]. SCFA are the major energy source for colonic epithelial cells and could potentiallypromote adiposity by contributing additional calories and by fueling into gluconeogenesis and lipogenesis in adipocytes or hepatocytes [37,38]. However, SCFA also have many beneficial effects on host physiology and therefore targeting SCFA has also been suggested as potential therapeutic approach for obesity [39]. SCFA play an important role in host immunity [40]. A reduction in fecal SCFA has been linked to inflammatory and metabolic diseases such as obesity, diabetes and inflammatory bowel disease (IBD) [41]. Butyrate in particular possesses anti-cancer and anti-inflammatory properties [42].”Likewise the heterogeneity is an important concept in personal nutrition, as the authors point out, there will be an effect of genetics.Host immunity is very important and this is not clear from lines 142 to 144. Wethank the reviewer for pointing out this important aspect.We added these considerationsto the conclusion.
The updated passage(lines 223-227) reads:“Developing a general broad-spectrum diet high in fermentable fiber and unsaturated fat that improves physiological parameters and metabolic health would be an ideal goal, as it represents a cost-effective way to improve public and individual health. However, this might not be feasible given the high interindividual dietary preferences and variation in the gut microbiome but also host genetics and immunity.”The endotoxin theory of inflammationand gut origin of obesity needs clarification, rather than linking this to IBD.Obesity and metainflammation is clearly an important clinical endpoint to treatment of diabetes and obesity and needs to be explained better.Wethank the reviewer for pointing out this important point.We expanded this section by adding a discussion on the endotoxin theory of inflammation.The updated passage(lines 116-121) reads:“In obesity, the concept of endotoxin-mediated low-grade inflammation bridges nutrition, microbiome and inflammation. According to this theory, Western diet-induced dysbiosis increases the level of endotoxin, a component of the outer membrane of Gram-negative bacteria, which then reaches other organs such as the liver and adipose tissues via the blood circulation in higher concentrations that suffice to stimulate tissue-resident immune cells and thereby lead to immune responses and impaired organ functions [45].”The use of germ-free mice with an immature immune system and the limits of using this as a model for humans needs to be emphasized. The receptors for SCFA, TLR4, and immunopathology of these animals is unlike that of humans.We added a short passage on the limitations of gnotobiotic animal models to the discussion.The updated passage(lines 201-209) reads:“Taken together, recent translational efforts began to transfer findings from exploratory and preclinical studies into functional human trials targeting specific functional host-microbiome interactions. This translation to humans is pivotal, as preclinical research models such as gnotobiotic animals on the one hand are essential tools to uncover functional principles in nutrition-host-microbiome interactions. On the other hand, germ-free animals also have an immature immune system anddisplay differences in important immune components and the composition of the microbiome, which limits their use as human surrogates. Nonetheless, an improved understanding of host-microbiome interactions from a variety of diseases led to the development of novel interventions that could improve traditional therapeutic concepts.”
